# Unsupervised Disentanglement with Tensor Product Representations on the Torus

**Michael Rotman[1], Amit Dekel[2], Shir Gur[1], Yaron Oz[3,4], Lior Wolf[1]**

[1] School of Computer Science, Tel-Aviv University, Israel

[2] Univrses, Sweden

[3] School of Physics and Astronomy, Tel-Aviv University, Israel

[4] Simons Center for Geometry and Physics, Stony Brook, USA

`rotmanmi@post.tau.ac.il`, `amit.dekel@univrses.com`, `shir.gur@cs.tau.ac.il`
`yaronoz@tauex.tau.ac.il`, `wolf@cs.tau.ac.il`

## Abstract

The current methods for learning representations with auto-encoders almost exclusively employ vectors as the latent representations. In this work, we propose to employ a tensor product structure for this purpose. This way, the obtained representations are naturally disentangled. In contrast to the conventional variations methods, which are targeted toward normally distributed features, the latent space in our representation is distributed uniformly over a set of unit circles. We argue that the torus structure of the latent space captures the generative factors effectively. We employ recent tools for measuring unsupervised disentanglement, and in an extensive set of experiments demonstrate the advantage of our method in terms of disentanglement, completeness, and informativeness. The code for our proposed method is available at `https://github.com/rotmanmi/Unsupervised-Disentanglement-Torus`.

## 1 Introduction

Unsupervised learning methods of disentangled representations attempt to recover the explanatory factors $z$ of variation in the data. The recovered representation, $c$, is expected to be: (i) *disentangled*, i.e., each element in $c$ should vary one of the generative factors in $z$ (ii) *complete* in the sense that all generative factors $z$ can be controlled by the latent representation $c$, and (iii) *informative*, that is the ability to relate via a low capacity model (e.g., a linear model) the generative factors $z$ and the latent representation $c$.

It has been shown that solving this task without further assumptions is impossible, since there is an infinite number of bijective functions $f$ that map between $c$ and $z = f(c)$ (Locatello et al., 2019) with all corresponding generative models having the same marginal distributions of the observations. In this work, we propose to use representations that are a tensor product of elements, which take values on unit circles $S^1$, claiming that this structure is suitable for the effective recovery of the underlying modes of variation.

This claim follows from entropy considerations. One wishes to have a representation that has a low *entropy of entanglement*. The entropy of entanglement is measured by evaluating the number of non-zero eigenvalues of the Schmidt decomposition of the representation. If a representation can be described by only one term, which is an outer product of the orthogonal basis functions, it has the lowest possible entanglement entropy, zero. If a representation requires the use of more than one term in the decomposition, it is entangled and its entanglement entropy can be quantified, e.g. by the Von Neumann entropy constructed from non-zero eigenvalues.

The tensor product of $n$ unit circles, $S^1$, takes the shape of an $n$-torus $T^n = (S^1)^n$, and has a low entanglement property. This representation has the advantage that it can capture the periodicity structure of the generative factors, where each circle controls a different aspect of the original distribution. Unlike other generative models, which rely on the Gaussian distribution as a prior, $T^n$ is

a compact manifold, and therefore any function that acts on it, such as a decoder, has to interpolate only, but not extrapolate when generating new instances.

In this work, we present the $T^D$-VAE, a Variational Auto Encoder whose latent space resides on the $T^D$ torus manifold. In an extensive set of experiments with four different datasets, we compare the torus latent representations with others proposed in the literature. We show a clear advantage of the torus representation in terms of various measures of disentanglement, completeness, informativeness and propose a new metric, the DC-score, which assesses the combined disentanglement and completeness performance. We present a detailed quantitative and qualitative analysis that supports our method.

## 2  RELATED WORK

Generative models aim to create new instances that are indistinguishable from a given distribution. Auto encoders (Kramer, 1991) achieve this by constructing the identity function using a composition of two components, an encoder and a decoder. The encoder is tasked with compressing the input into a latent representation with a dimension smaller than the input's, whereas the decoder is responsible for reconstructing the original input from the latent space.

Such auto encoders usually overfit to their training data and are very limited when tasked with generating new samples. Instead of mapping each input instance to a constant vector, Variational Auto Encoders (Kingma & Welling, 2013) (VAE) map each instance to a pre-defined distribution, by minimizing both the encoder-decoder reconstruction loss and a KL-divergence term that depends on the prior distribution. VAE's are capable of generating new instances by sampling a latent vector from the predefined distribution; however, the interpretation of latent space components is typically obscure and the mapping between them and the dataset properties may be complex.

In order to increase the latent space interoperability, various modifications of the original VAE were introduced. These variants try to achieve a factorization of the latent space components with each component corresponding to a specific simple characteristic of the dataset. In $\beta$-VAE (Higgins et al., 2016), the weight of the KL-divergence term is increased w.r.t. the reconstruction term, which results in better factorization of the latent space. However, this introduces a new hyper-parameter and departs from the theoretical derivation of the ELBO term. For example, large $\beta$ parameters may prevent the conditional distribution from modeling the data faithfully. DIP-VAE (Kumar et al., 2017) introduces a regularizer that constrains the covariance matrix of the posterior distribution. This term is added to the original VAE loss function, along with two hyper-parameters, and helps achieve better factorization. Factor-VAE (Kim & Mnih, 2018) adds a Total Correlation (TC) approximation penalty over the produced latent codes and introduces an additional hyper-parameter.

### 2.1  DISENTANGLEMENT METRICS

In order to evaluate the expressiveness of different generative models, a plethora of disentanglement definitions and metrics were suggested in the literature (Do & Tran, 2019; Eastwood & Williams, 2018), see Locatello et al. (2019) for a comprehensive review. In this work, we adopt the definitions and metrics introduced in Eastwood & Williams (2018) which provides a successful set of disentanglement metrics.

The authors of Eastwood & Williams (2018) introduce three metrics, *Disentanglement*, *Completeness* and *Informativeness* (DCI), to quantify the disentanglement properties of the latent space and its ability to characterize the generative factors of a dataset. Given a dataset that was generated using a set of $K$ generative factors, $\vec{z} \in \mathbb{R}^K$, and that it is required to learn latent codes, $\vec{c} \in \mathbb{R}^D$ of $D$-dimensional representation. Ideally, in an interpretable representation, each generative factor $z_i$ would correspond to only one latent code $c_a$. It is also beneficial if the mapping is linear, as the correlation between generative factors and codes can then be easily assessed. Generating new instances that are indistinguishable from the original dataset distribution requires that the latent codes cover the whole range of the generative factors. Furthermore, such a representation provides the ability to modify a specific property of a generated instance by directly tuning the corresponding latent code.

The DCI metrics aim to quantify the relationship between codes and generating factors by having a single number that characterizes the relative importance of each code,[1] $c_a$, in predicting a factor, $z_i$, which in turn defines an importance matrix $R_{ai}$. To construct the importance matrix, $K$ regressors are trained to find a mapping between $z_i$ and $\vec{c}$, $\hat{z}_i = f_i(\vec{c})$. In this work, we follow Eastwood & Williams (2018) and infer the importance matrix using a lasso linear regressor's weights, $W_{ia}$, by $R_{ia} = |W_{ia}|$.

Once the importance matrix $R_{ai}$ is obtained, the DCI metrics can be defined explicitly. The **disentanglement** is given by

$$\mathcal{D} = \frac{\text{rank}\,(R)}{K} \sum_{a=1}^{D} \rho_a \mathcal{D}_a \,, \tag{1}$$

where

$$\mathcal{D}_a = 1 - H_K(P_a) = 1 + \sum_{k=1}^{K} P_{ak} \log_K P_{ak}, \quad P_{aj} = \frac{R_{aj}}{\sum_{k=1}^{K} R_{ak}}, \quad \rho_a = \frac{\sum_j R_{aj}}{\sum_{bk} R_{bk}} \,. \tag{2}$$

High disentanglement means that each entry of the latent vector $\vec{c}$ corresponds to only one element in $\vec{z}$ (in a linear sense), that is, each code element, $c_a$, affects one generating factor. The disentanglement metric defined above differs from Eastwood & Williams (2018) by a correction factor $\frac{\text{rank}(R)}{K}$. When the rank of $R$ is equal to the generative factor's dimension, this correction equals 1, and does not affect the metric. However, it does make a difference when the number of codes is smaller than what is needed to account for all generating factors. Typically, one assumes that there are at least as many expressive codes as the number of factors. When this is not the case, it means that even if we have disentanglement among the expressive codes, their number is not sufficient, and our correction accounts for this. Note that the correction has no influence if there are irrelevant code elements in addition to a sufficient number of expressive ones. The $\rho$ factors handle cases where some of the code dependence on the factors is very weak (namely irrelevant codes), while other codes depend mainly on one factor and hence should not be of equal importance.

The **completeness** is defined by

$$\mathcal{C} = \frac{1}{K} \sum_{j=1}^{K} \mathcal{C}_j \,, \quad \mathcal{C}_j = 1 - H_D(\tilde{P}_j) = 1 + \sum_{a=1}^{D} \tilde{P}_{aj} \log_D \tilde{P}_{aj} \,, \quad \tilde{P}_{aj} = \frac{R_{aj}}{\sum_{b=1}^{D} R_{bj}} \,. \tag{3}$$

In contrast to the disentanglement metric, which considers a weighted mean over the generative factors, motivated by the fact that irrelevant units in $\vec{c}$ should be ignored, here all codes are treated equally.

A situation in which each factor $z_i$ is explained by only one element of the code $c_a$ results in high completeness.

The **informativeness** is the MSE between the ground-truth factors, $z$, and the predicted values, $f_j(\vec{c})$,

$$\mathcal{I} = \frac{1}{K} \sum_{j=1}^{K} \mathbb{E}_{\text{dataset}} \left[ |z_j - f_j(\vec{c})|^2 \right] \,. \tag{4}$$

Disentanglement and completeness values range between zero and one, where one is the best value, while the best value for informativeness is zero.

Disentanglement and completeness alone do not provide sufficient information on whether a representation factorizes properly. For example, consider the case of a representation where only one generative factor is described by all the codes. While this representation is completely disentangled, it is not complete, and is therefore meaningless. In order to have a meaningful representation, both disentanglement and completeness need to be high. Thus, we introduce a new score, called the DC-score, which accounts for both disentanglement and completeness. We define the DC-score as the geometric mean of the two metrics, DC-score $= \sqrt{\mathcal{D}\mathcal{C}}$. This way, only cases where both scores are high will result in a high DC-score. Furthermore, the score will favor cases where both disentanglement and completeness are comparable rather than having different values while having the same arithmetic mean.

---

[1] throughout the paper we use $a, b, c, \ldots$ and $i, j, k, \ldots$ letters for indices of the codes $\vec{c}$ and the factors $\vec{z}$ components respectively.

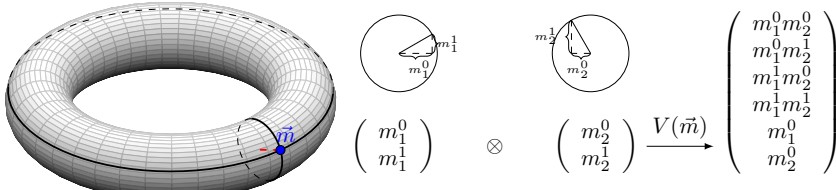

Figure 1: An example of how a tensor product of two circles, $T^2 = S^1 \times S^1$, is converted to a flattened latent representation that can be fed into the decoder. Each point on the torus can be described by two angles. Each angle represents an $m_a$ tuple. The tensor product of the $m_a$ tuples is concatenated with their respective $\alpha = 0$ component to produce the input to the decoder.

## 3 METHOD

Assume a pre-determined set of vectors sampled from an unknown distribution $\chi$, each vector containing $K$ independent factors, $\vec{z} \in \mathbb{R}^K$. The training dataset contains samples $x_I = F(\vec{z}_I)$, $I = 1, \ldots, N$ that are generated using a generative function $F$. In unsupervised disentanglement, the learner has no access to $\vec{z}$ nor to $F$, and receives only the set of samples, $\{x_I\}_{I=1}^N$.

The learner has to recover a set of representation vectors $\vec{c}_I$ that are linked to $\vec{z}_I$ in a way that is bijective and disentangled, see Section 2.1. Furthermore, in order to generate new samples, it is also necessary to be able to sample new representation vectors $\vec{c}_{\text{new}}$, and to obtain a generative function $G$ such that the new generated samples are from the same distribution of $F(\vec{z})$, where $\vec{z} \sim \chi$.

In this work, we view the latent representation vector $\vec{c}$ as the angles associated with a list of $D$ two-dimensional unit vectors $m_a \in \mathbb{R}^2$, $1 \leq a \leq D$, i.e., $\forall a, \|m_a\| = 1$. Using theses two-dimensional vectors we define

$$v_{prod} \equiv \text{vec}\left(v_{prod}^{\alpha_1 \ldots \alpha_D}\right), \quad v_{prod}^{\alpha_1 \ldots \alpha_D} = m_1^{\alpha_1} \otimes \cdots \otimes m_D^{\alpha_D}, \quad v_{orient} \equiv \left(m_1^0, \ldots, m_D^0\right), \quad (5)$$

where $\otimes$ is the outer product operator, $\alpha_a \in \{0, 1\}$ and vec is the vectorization operation (equivalent to flattening the tensor). Define the operator $V$, that given $m_1, \ldots, m_D$, concatenates $v_{prod}$ together with $v_{orient}$, $V(m_1, \ldots, m_D) = [v_{prod}; v_{orient}]$. Fig. 1 depicts the case for $D = 2$.

The vector $v = V(m_1, \ldots, m_D)$ resides in a vector subspace of $\mathbb{R}^{2^D + D}$, defined by only a set of $D$ parameters. The additional $D$ elements of $v_{orient}$ are required to ensure that the mapping $V$ is bijective. For example, this can be seen by examining the case of $D = 2$: let $m_1 \equiv (\cos\theta_1, \sin\theta_1)$, $m_2 \equiv (\cos\theta_2, \sin\theta_2)$, $m_1' \equiv -m_1$ and $m_2' \equiv -m_2$. Then $m_1 \otimes m_2 = m_1' \otimes m_2'$.

A natural way of acquiring a random point on the circle, $S^1$, is by sampling two independent Gaussians, $\hat{m}_a^{\alpha_k} \sim \mathcal{N}(\mu_a^{\alpha_k}, \sigma_a^{\alpha_k})$. Each tuple of these vectors is then normalized to have a unit norm,

$$m_a^{\alpha_k} = \frac{\hat{m}_a^{\alpha_k}}{\sqrt{(\hat{m}_a^0)^2 + (\hat{m}_a^1)^2}} . \quad (6)$$

Assume that the elements, $\hat{m}_a^{\alpha_k}$, follow the normal distribution with a zero mean and a standard deviation of 1, $\hat{m}_a^{\alpha_k} \sim \mathcal{N}(0, 1)$, then the vectors $m_a$ follow the uniform distribution on $S^1$.

For the purpose of obtaining the distribution parameters of $\hat{m}_a^{\alpha_k}$, the encoder $e$ is applied on an instance, $x$,

$$e(x) = \begin{bmatrix} \mu_1^0, & \sigma_1^0 \\ \mu_1^1, & \sigma_1^1 \end{bmatrix} \cdots \begin{bmatrix} \mu_D^0, & \sigma_D^0 \\ \mu_D^1, & \sigma_D^1 \end{bmatrix} . \quad (7)$$

The reparametization trick (Kingma & Welling, 2013) is then used to obtain a set of normally distributed vectors. Denote by $S$ the sampling operator, we sample the coding vector as

$$\vec{M}_I \equiv \left\{ \begin{pmatrix} m_1^0 \\ m_1^1 \end{pmatrix}, \ldots, \begin{pmatrix} m_D^0 \\ m_D^1 \end{pmatrix} \right\} = S(e(x_I)), \quad (8)$$

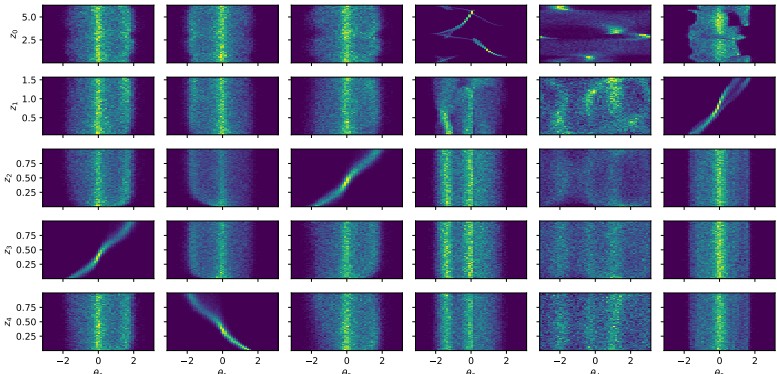

Figure 2: Heatmap of the generative factors vs. the codes for the teapots dataset with our architecture with six circles. The factors are ordered by azimuth and elevation angles, followed by RGB values. The disentanglement, completeness and DC-scores are 0.67, 0.59 and 0.64, respectively. One can clearly see the high disentanglement property where each code mostly controls only one factor and high completeness where most factors are controlled by one code (e.g. $\theta_0$ controlling $z_3$).

where each component, $(a, \alpha_a)$ of $m$, $a = 1 \ldots D$, $\alpha = 0, 1$ is sampled from $\mathcal{N}(\mu_a^\alpha, \sigma_a^\alpha)$.

The decoder $G$ then acts on $V(m_1, \ldots, m_D)$ to generate a new instance, $\tilde{x}$. The requirement that the generated instance is indistinguishable from the original distribution $F(\chi)$ translates to minimizing the ELBO loss function. First, it maximizes the log-likelihood probability of $\tilde{x}$ to be similar to the training sample it reconstructs by having an l2 loss term. Second, to encourage a uniform distribution on the torus, it contains a KL-divergence term with respect to $\mathcal{N}(0, \mathbb{1}_D)$. The overall expression for the loss function used during the optimization of the networks $e$ and $G$ is

$$\mathcal{L} = \sum_I \left[ \left| G\left( V\left( \vec{M}_I \right) \right) - x_I \right|^2 - \beta D_{KL}\left( p\left( \hat{m} \right) || r_I \right) \right], \tag{9}$$

where $\beta$ is a hyperparameter introduced in $\beta$-VAE (Higgins et al., 2016) and $r_I \sim \mathcal{N}(0, \mathbb{1}_D)$.

As the latent representation resides on the torus, sub-vectors $2D$ components of $m_a$ may be described by $D$ angles, $\theta_a \in [0, 2\pi]$ (the angles $\theta_a$ are identified with the codes $c_a$). In order to generate new instances, identify, $m_a^0 = \cos\theta_a$, $m_a^1 = \sin\theta_a$. and apply the decoder $G(V(m))$ to these elements.

The torus topology, $T^D$, is both compact and periodic. Since it is compact, for every sampled $\theta_a$, there is a nearby $\theta_a'$ from the training set, thus, the network $G$ can be viewed as an interpolator. This is in contrast to vectors sampled over $\mathbb{R}^D$, in which $G$ has to extrapolate. Furthermore, being periodic, this topology is able to exploit a periodic structure in the generative factors. Consider a common generative factor associated with rotations. A rotation cannot be represented by a single non-compact variable; therefore encoding a rotation in a latent space requires the entanglement of two components. However, on the torus, only one compact dimension can be used to identify this generative factor.

## 4  EXPERIMENTS

We compare our method to the leading VAE architectures from the current literature. The baseline methods are (i) $\beta$-VAE (Higgins et al., 2016), (ii) DIP-VAE-II (Kumar et al., 2017), and (iii) Factor-VAE (Kim & Mnih, 2018). The code for all methods was taken from the Pytorch-VAE repository (Subramanian, 2020) (Apache License 2.0). The network architectures for both the encoder, decoder and the Factor-VAE's discriminator (which we do not use) follow those in Eastwood & Williams (2018), see Table 1. Each horizontal line in Table 1 denotes a skip-connection.

All methods were trained using ADAM (Kingma & Ba, 2014) with a learning rate of 0.0001 and a batch size of 144. The hyperparameters used for DIP-VAE were $\lambda_{\text{diag}} = 10$ and $\lambda_{\text{off-diag}} = 1.0$. All methods were trained for a maximum number of 50 epochs, and the metrics were evaluated

using the model that performs best on the validation set with respect to MSE. Furthermore, each baseline method was evaluated using two latent representations sizes, $D = 10, 128$ for all datasets. For our method, $T^8$-VAE, we use 8 circles, $\left(S^1\right)^8$. This results in $V \in \mathbb{R}^{136}$, leaving the number of parameters in the decoder comparable to the other baseline methods. Each experiment was evaluated with five different seeds and the mean value is reported.

In order to compute the DCI metrics, we used a linear lasso regressor based on the implementation of Eastwood & Williams (2018) (MIT License) and Zaidi et al. (2020) (Apache License 2.0). For each generative factor, a lasso regressor fits between the normalized codes obtained by the encoder, $e$, to a normalized factor, namely $f_i(\vec{c})$ for each $z_i$. For each factor, the $\alpha$ parameter for the regressor using 10-fold cross-validation from the set $\left\{10^{-6}, 10^{-5}, 10^{-4}, 10^{-3}, 10^{-2}, 0.1, 0.2, 0.4, 0.8, 1\right\}$, picking the one with the minimal the MSE error. The weights of the lasso regressor are then used to construct the $W \in \mathbb{R}^{K \times D}$ matrix from which the metrics are computed as explained in section 2.1.

The Fréchet Inception Distance (Heusel et al., 2017) (FID) is used to assess the quality of image generation. Specifically, we use the implementation of Seitzer (2020) (Apache License 2.0) and report the FID at the final average pooling features of the Inception3 NN (Szegedy et al., 2016).

**The teapots dataset** (Moreno et al., 2016) contains $200,000$ $64 \times 64$ images of camera-centered teapots rendered using a set of 5 generative factors, azimuth that is sampled from $\sim U\left[0, 2\pi\right)$, elevation that is sampled from $\sim U\left[0, \frac{\pi}{2}\right)$ and three RGB channels, each sampled from $U\left[0, 1\right]$. All VAEs are trained over a subset of $160,000$ (the generative factors are unknown to all architectures), whereas the disentanglement metrics are evaluated on the remaining $40,000$ images.

**The Cars3D dataset** (Reed et al., 2015) contains $183$ CAD models of different cars projected into $64 \times 64$ image with four elevations and $24$ azimuths. These three generative factors are split between a training set of $14054$ images and a validation set of $3514$ images.

**The 2dshapes Dataset**, introduced here and released to the public (MIT License), contains $200,000$ $64 \times 64$ images of four possible shapes, triangles, squares, pentagons and hexagons. All shapes in the dataset are centered in the image and have a white background. The factors controlling the images are the following: specific shape drawn from a discrete uniform distribution, a scale value between $s \sim U[20, 40]$, rotation angle $\theta \sim U[0, 2\pi]$, and color RGB values $r \sim U[0, 1]$, $g \sim U[0, 1]$, $b \sim U[0, 1]$ all sampled uniformly. Since the shapes are symmetric, different angles could correspond to the same image. The training set contains a subset of $160,000$ instances, whereas the disentanglement metrics are evaluated on the remaining $40,000$ validation instances.

**The 3dshapes dataset** (Burgess & Kim, 2018) contains $480,000$ $64 \times 64$ images of four possible objects, a cylinder, a tube, a sphere and a box rendered with 8 linearly spaced scale values, 10 hues linearly spaced, floor wall and object hue values and fifteen different orientations, which amounts to six generative factors.

**The dSprites Dataset** (Matthey et al., 2017) contains $737,280$ $64 \times 64$ binary images of three shapes, a square, an ellipse and a heart with six different, equally spaced scale values, 40 different orientations, and $32 \times 32$ possible positions in the plane, amounting to five generative factors. $589,824$ instances were used for training and evaluation was performed on the remaining $147,456$ validation instances.

**Results**     Across all the datasets (Tables 2), our $T^8$-VAE architecture obtains the highest DC-score, which combines the disentanglement and completeness scores, compared with the baselines. In all of these cases the generated images are meaningful, as can be seen from the relatively low FID scores. Furthermore, for teapots, 2dshapes and dSprites, we also achieve the best reconstruction MSE and FID scores. In the 3dshapes dataset we obtain the best MSE score and comparable FID scores to $\beta$-VAE, which achieves the highest score. Qualitative results for the influence of each circle for the teapots and 2dshapes datasets can be seen in Fig. 3.

For the Cars3D dataset the results are quite different, even though our DC-score is highest. The best performing method in terms of the MSE and FID scores is DIP-VAE-II 128, which is not among the best performing methods for the other datasets. One plausible reason is that this dataset, although having only three generative factors, hides a much richer high-dimensional distribution. The factors controlling the car model (183 in total) implicitly contain many characteristics of a car, which requires a high dimensional latent space. Nonetheless, our method achieves the second best FID score for this dataset.

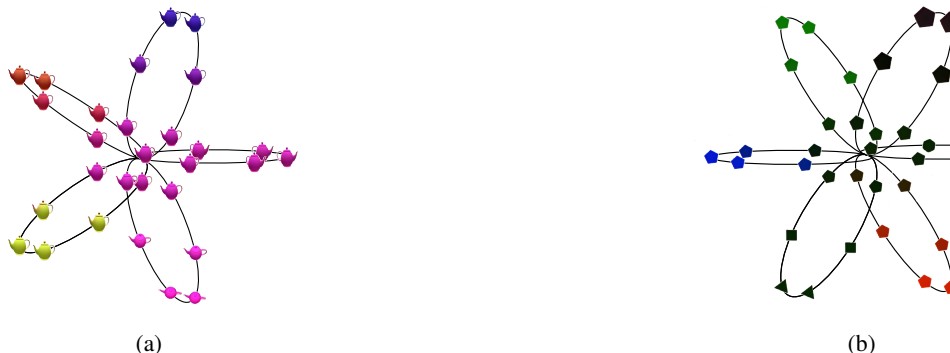

(a)                (b)

Figure 3: The latent representation of the $T^8$-VAE. Each $S^1$ circle corresponds to a different factor of the dataset that is varied as the angle changes. (a) The 5 most expressive circles on the Teapot dataset. (b) The 6 most expressive circles on the 2dshapes dataset.

Table 1: Architectures for the encoder $e$, the decoder $G$ and the discriminator used for Factor-VAE. These architectures follow the ones in Eastwood & Williams (2018).

| Encoder | Decoder | Discriminator |
|---|---|---|
| $3 \times 3$ 64 conv | FC $d \to 8192$ | FC $d \to 1000$ |
| BN, ReLU, $3 \times 3$ 64 conv
BN, ReLU, $3 \times 3$ 128 conv | BN, ReLU, $3 \times 3$ 512 conv
BN, ReLU, $3 \times 3$ 512 conv | BN, LeakyReLU 0.2
FC $1000 \to 1000$ |
| BN, ReLU, $3 \times 3$ 128 conv
BN, ReLU, $3 \times 3$ 256 conv | BN, ReLU, $3 \times 3$ 256 conv
BN, ReLU, $3 \times 3$ 256 conv | BN, LeakyReLU 0.2
FC $1000 \to 1000$ |
| BN, ReLU, $3 \times 3$ 256 conv
BN, ReLU, $3 \times 3$ 512 conv | BN, ReLU, $3 \times 3$ 128 conv
BN, ReLU, $3 \times 3$ 128 conv | BN, LeakyReLU 0.2
FC $1000 \to 2$ |
| BN, ReLU, $3 \times 3$ 512 conv
BN, ReLU, $3 \times 3$ 512 conv | BN, ReLU, $3 \times 3$ 64 conv
BN, ReLU, $3 \times 3$ 64 conv | |
| FC $8192 \to d$ | BN, ReLU, $3 \times 3$ 3 conv, tanh | |

**Dependence on $\beta$ and on $D$**    In order to examine the sensitivity of our method to different tori dimensions, as well as the influence of the $\beta$ parameter on KL-divergence, we examined the DC-score and reconstruction error on the Teapot dataset, using four topologies, $T^4$, $T^5$, $T^6$ and $T^8$, and using five different $\beta$ coefficients, $0, 1, 3, 6, 9$ ($0$ corresponds to no KL-divergence). As can be seen in Fig. 4, removing the KL-divergence completely, leads to a decrease in the reconstruction error; however, the DC-score also decreases. The $T^4$-VAE exhibits the worst reconstruction error - not surprisingly, as it can only encode four generative factors, whereas there are five generative factors in the dataset. Both the $T^5$-VAE, $T^6$-VAE and $T^8$-VAE have the capacity of encoding all generative factors, however, the encoder of the $T^8$-VAE should ignore three of the circles whereas the encoders of the $T^5$-VAE and $T^6$-VAE should ignore at most one circle. Thus, when increasing $\beta$, the induced angles on $T^8$ tend to a more uniform distribution, which entangles between themselves. This does not happen in $T^5$-VAE and $T^6$-VAE, since there is at most one extra component that can be entangled.

**Heatmap analysis** Disentanglement results can be qualitatively evaluated through a visual inspection of the 2d heatmaps of the generative factors $z_i$ vs. the codes $c_a$ (the angles $\theta_a$ in our case). In Figure 2 we show an example of the heatmap results for using our method with $D = 6$, namely using a $T^6$-VAE. The disentanglement and completeness properties are clearly seen in the $z_1, z_2, z_3, z_4$ dependence on $\theta_5, \theta_2, \theta_0, \theta_1$ respectively, while we see worse completeness for the $z_0$ factor. Such images help us better understand cases where one of the scores is high and the other is low, or whether a specific regressor is suited for capturing the functional dependence.

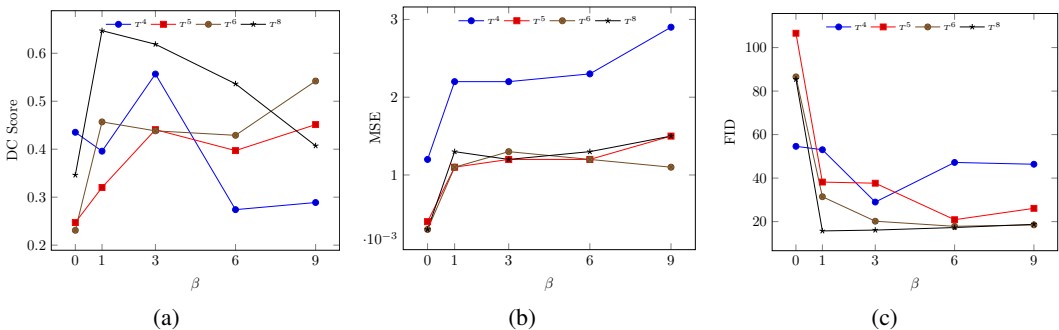

Figure 4: The performance dependence on the latent torus and $\beta$ parameter of the KL-divergence term with five generating factors.(a) The accumulated DC-score. (b) The reconstruction error. (c) The FID score. We see a clear advantage for a latent space of at least the same dimension as the number of generating factors.

Table 2: Results on all the datasets.

| Dataset | Method | DC-Score | $\mathcal{D}$ | $\mathcal{C}$ | $\mathcal{I}$ | MSE | FID |
|---|---|---|---|---|---|---|---|
| teapots | $\beta$-VAE 10 | 0.1796 | 0.1902 | 0.1702 | 0.0263 | 0.0178 | 154.9922 |
| | $\beta$-VAE 128 | 0.1286 | 0.1715 | 0.0964 | **0.0123** | 0.0187 | 159.7412 |
| | DIP-VAE-II 10 | 0.1226 | 0.2480 | 0.0676 | 0.0534 | 0.0186 | 151.7267 |
| | DIP-VAE-II 128 | 0.0966 | 0.1592 | 0.0593 | 0.0180 | 0.0063 | 62.8249 |
| | Factor-VAE 10 | 0.2526 | 0.2183 | 0.3119 | 0.0601 | 0.0167 | 110.7673 |
| | Factor-VAE 128 | 0.3891 | 0.3564 | 0.4761 | 0.0586 | 0.0061 | 70.9153 |
| | $T^8$-VAE | **0.6230** | **0.6558** | **0.5919** | 0.0332 | **0.0012** | **15.7170** |
| 2d shapes | $\beta$-VAE 10 | 0.2898 | 0.2913 | 0.2883 | 0.0335 | 0.0145 | 143.7522 |
| | $\beta$-VAE 128 | 0.1733 | 0.2748 | 0.1093 | **0.0216** | 0.0165 | 178.0128 |
| | DIP-VAE-II 10 | 0.1701 | 0.2525 | 0.1342 | 0.0650 | 0.0224 | 221.5138 |
| | DIP-VAE-II 128 | 0.1125 | 0.2354 | 0.0543 | 0.0378 | 0.0131 | 78.1923 |
| | Factor-VAE 10 | 0.1875 | 0.2104 | 0.1676 | 0.0660 | 0.0214 | 122.2750 |
| | Factor-VAE 128 | 0.5165 | 0.3935 | **0.6847** | 0.0844 | 0.0062 | 86.3864 |
| | $T^8$-VAE | **0.7021** | **0.7274** | 0.6777 | 0.0321 | **0.0009** | **17.3992** |
| 3dShapes | $\beta$-VAE 10 | 0.5121 | 0.5319 | 0.4934 | 0.0420 | 0.0020 | **18.3127** |
| | $\beta$-VAE 128 | 0.1219 | 0.1739 | 0.0854 | **0.0070** | 0.0023 | 20.0576 |
| | DIP-VAE-II 10 | 0.0838 | 0.2211 | 0.0329 | 0.0867 | 0.0046 | 186.8153 |
| | DIP-VAE-II 128 | 0.1052 | 0.1799 | 0.0617 | 0.0185 | 0.0115 | 267.8511 |
| | Factor-VAE 10 | 0.1759 | 0.2195 | 0.1418 | 0.0738 | 0.3748 | 139.5286 |
| | Factor-VAE 128 | 0.1884 | 0.2166 | 0.1641 | 0.0236 | 0.0024 | 62.3407 |
| | $T^8$-VAE | **0.7112** | **0.8301** | **0.6140** | 0.0731 | **0.0009** | 22.2525 |
| Cars3D | $\beta$-VAE 10 | 0.1962 | 0.2917 | 0.1321 | 0.0897 | 0.1152 | 393.0866 |
| | $\beta$-VAE 128 | 0.1447 | 0.2469 | 0.0848 | 0.0709 | 0.1153 | 384.6031 |
| | DIP-VAE-II 10 | 0.2090 | 0.3979 | 0.1106 | 0.0789 | 0.0629 | 259.0138 |
| | DIP-VAE-II 128 | 0.1226 | 0.2261 | 0.0665 | **0.0492** | **0.0206** | **191.8091** |
| | Factor-VAE 10 | 0.2551 | 0.3800 | 0.1729 | 0.0961 | 0.0817 | 274.6059 |
| | Factor-VAE 128 | 0.3630 | **0.6047** | 0.2351 | 0.0892 | 0.0738 | 315.6645 |
| | $T^8$-VAE | **0.4623** | 0.5984 | **0.3571** | 0.0783 | 0.0465 | 242.2127 |
| dSprites | $\beta$-VAE 10 | 0.2047 | 0.2600 | 0.1613 | 0.0843 | 0.0116 | 77.3847 |
| | $\beta$-VAE 128 | 0.1160 | 0.1837 | 0.0733 | **0.0548** | 0.0125 | 77.4040 |
| | DIP-VAE-II 10 | 0.1886 | 0.2673 | 0.1335 | 0.0897 | 0.0209 | 134.6068 |
| | DIP-VAE-II 128 | 0.1065 | 0.2237 | 0.0509 | 0.0645 | 0.0218 | 168.3634 |
| | Factor-VAE 10 | 0.3020 | 0.2619 | **0.3908** | 0.0932 | 0.1529 | 153.1589 |
| | Factor-VAE 128 | 0.1654 | 0.2373 | 0.1154 | 0.0571 | 0.0419 | 101.6020 |
| | $T^8$-VAE | **0.3396** | **0.3681** | 0.3161 | 0.0767 | **0.0066** | **65.7015** |

## 5 DISCUSSION

We proposed and analyzed a low entangled latent representation of the generating factors of a distribution. The latent space has the topology of a torus which is a direct product of circles. Each circle controls one generative factor, and moving along the circle changes the corresponding aspect of the distribution. Our latent space construction resembles the phase space structure of a classical, integrable physical system, whose dynamics can be described using action-angle canonical variables as motion on a phase space torus. The dimension of the torus is the number of the system's degrees of freedom, and each point on it specifies the location in space and the velocity of the system at a given instant in time. The coordinates of the point on each circle correspond to a particular periodic cycle to which the whole motion has been decomposed. Analogously, the coordinates on each circle of a point in the latent space torus specify one of the basic properties of an element of the dataset to which the description of that element has been decomposed. The possible intriguing relationship between the integrable structure of dynamical systems and its decomposition into its basic harmonic motions and the decomposition of the dataset into its basic constituents can be valuable in relating dynamical systems to the process of learning.

The task of unsupervised disentanglement learning is complicated by the fact that given any disentanglement representation there is an infinite family of entangled representations that have the same marginal distribution (Locatello et al., 2019). Therefore, an inductive bias is needed for the task. Our solution in this work is based on the concept of entanglement entropy in quantum physics. There are non-generic states in the Hilbert space that have low entanglement, and a special class of them are the product states, whose entanglement entropies are identically zero. We employed these states for our disentanglement representation. It is indeed easy to see that these provide a special representation, since any random unitary (or orthogonal) matrix acting on them will make our representation entangled and hence destroy the disentanglement properties of our solution. Thus, in a sense, our inductive bias is the use of the special non-generic states (of measure zero) in the Hilbert space for our representation.

We introduced a DC-score to jointly quantify the disentanglement and completeness of the torus latent representation and calculated it, as well as various other measures in a large set of numerical experiments. For the comparison we used five different types of datasets. We varied the dimension of the latent space torus and showed a clear advantage of the torus latent description compared to others when the torus dimension is equal or greater than the number of generating factors. In order to properly assess the cases where the number of expressive codes is lower than the number of generative factors, we introduced a correction to the disentanglement score, and hence to our DC-score.

One limitation of our proposed low entangled representation is in scaling to datasets containing a large number of generative factors. The dimensionality of $v$ increases exponentially with the dimension of the torus latent representation and is, therefore, limited in describing these situations. This can perhaps be remedied by a different choice of embedding, e.g. working with a direct sum of tensor products, which increases polynomially. Another limitation of the model may be in its ability to effectively describe a property of the dataset that is not naturally encoded in a circle, but is, rather, non-compact in nature.

While our torus latent representation performs better compared to other approaches, one may still wonder why it did not achieve the perfect DC-score 1. This can be explained by the fact that the generation of the datasets assumes a certain number of generating factors, while in fact there could be more, some indirectly combined. For example, in a case where multiple underlying factors are combined into one explanatory factor, the disentanglement may overshoot, thereby reducing the obtained score.

## REPRODUCIBILITY STATEMENT

All the code for reproducing all experiments presented in this work is available in the supplement.

## ACKNOWLEDGMENTS

This project has received funding from the European Research Council (ERC) under the European Unions Horizon 2020 research and innovation program (grant ERC CoG 725974). The work of Y.O. is supported in part by the Israeli Science Foundation Center of Excellence. The contribution of the first author is part of a Ph.D. thesis research conducted at TAU.

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

## A  GENERATIVE FACTORS HEATMAPS

In this section we provide additional heatmap examples for different tori which provide a qualitative insight. Figures 5 and 6 present the heatmaps for our method on $T^4$ and $T^8$ torus topologies, respectively, with their $\mathcal{D}, \mathcal{C}$ and DC scores.

When the number of generative factors is larger than the number of torus dimensions, $D$, one can expect some mixing between the generative factors and the latent codes. This can be observed for instance for $T^4$ in Fig. 5.

As the dimension of the latent codes increases, see Fig. 6, the $\mathcal{D}, \mathcal{C}$ and $DC$-score metrics improve. This effect on the disentanglement results from the prefactor $\frac{rank(R)}{K}$ in equation 1, which depends on the ratio of the numbers of latent codes and generative factors when the latter is larger.

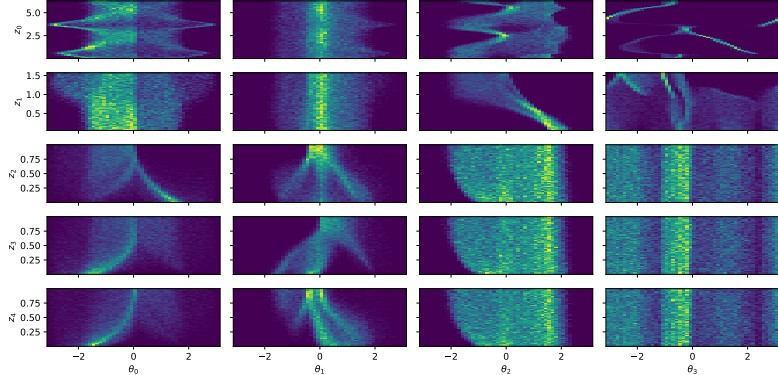

Figure 5: Heatmap of the generative factors vs. the codes for the teapot dataset with our architecture with four circles ($T^4$). The disentanglement, completeness and DC-scores are 0.36, 0.43 and 0.39, respectively. One can observe, for example that the second code $\theta_1$ controls all three RGB values.

## B  ON THE CONNECTION TO QUANTUM PHYSICS

The assumption is there is a set of elementary generative factors $z_i, i = 1, .., n$ that are disentangled from each other, and from which the data distribution can be constructed. The task is to learn these elementary factors and represent them by latent codes $c_a, a = 1, ..., m$. Ideally we would have liked to have $n = m$ and a bijective map $f$ that associates to any $c_a$ one $z_i$ and vice versa.

We are working with a linear representation and since quantum mechanics is a linear theory it inspires our choice of the latent space representation. In quantum mechanics states are represented by vectors in Hilbert space and the operators that act on them are linear. In our case, the generative factors constitute a state and so do the latent codes. The linear regressor relating them is a linear operator.

We represent each elementary factor $z_i$ as a state (qubit) in a Hilbert space $|z_i\rangle \in \mathcal{F}$ and the complete set of elementary factors by a state (quantum register of $n$ qubits) in the tensor product of these

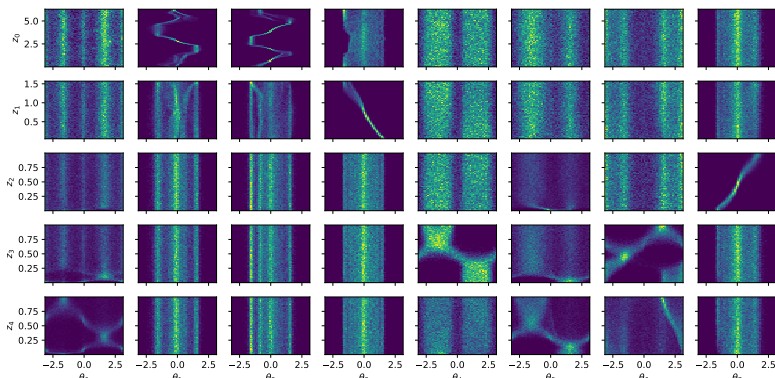

Figure 6: Heatmap of the generative factors vs. the codes for the teapot dataset with our architecture with eight circles ($T^8$). The disentanglement, completeness and DC-scores are 0.69, 0.61 and 0.65, respectively.

Hilbert spaces $|z_1, ..., z_n\rangle = |z_1\rangle \cdots |z_n\rangle \in \mathcal{H} = F \otimes F \cdots \otimes F$. Note, that only a measure zero of $n$-qubit states can be represented in such a way as a product state and the reason that we can do that is the assumption that $z_i$ are disentangled.

Entanglement can be viewed as a measure of how far the state is from being a product state. One way to measure it is by taking the supremum on the absolute value of the inner product of the state with all the possible product states. Another measure is the Von-Neumann entropy that is calculated by constructing a density matrix $\rho$ as the outer product of $|z_1, ..., z_n\rangle$, decomposing $z_i$ to e.g. two subsets $A$ and $B$ such that $\rho \in \mathcal{H}^A \otimes \mathcal{H}^B$ and tracing out part $B$, $\rho^A = Tr_B\rho$. The eigenvalues of $\rho^A$ encode the entanglement between $A$ and $B$. Specifically, the Von-Neumann entropy is $S_{VN} = -Tr\rho^A \log \rho^A$ and it vanishes iff the state $|z_1, ..., z_n\rangle$ is a product state.

Returning to our case, we construct the latent space representation using a tensor product since it should capture the above structure of the generative factors. The measure of disentanglement $\mathcal{D}_a$ quantifies it by the $-P \log P$ term which is the analog of the Von-Neumann entropy.

Note that there is a difference between our case and quantum mechanics: in quantum mechanics an elementary qubit state is represented by two angles $(\theta, \phi)$ while in our case and elementary latent code is one angle parametrizing a circle of the torus.

One may inquire what would have gone wrong in our analysis if we chose that space of latent codes to be another curved compact manifold such as a sphere. In such a case we should have covered the manifold with local patches such that there would be a linear map between the generative factors and the latent codes on each patch and a transition function between them. Such a representation would be entangled and will fail, e.g. on the sphere near the pole since rotating by the azimuthal angle would not change any property.

