# OpenReview forum: "Unsupervised Disentanglement with Tensor Product Representations on the Torus"
_ICLR.cc/2022/Conference — ICLR 2022 Poster_

### Official Review · Reviewer_UzjD · 2021-10-30

**Correctness:** 3
**Technical Novelty And Significance:** 2
**Empirical Novelty And Significance:** 2
**Recommendation:** 3
**Confidence:** 5

**Main Review:**

Strengths:
The method proposed is simple and looks easy to reimplement.

Weaknesses:
1. The motivation 'entropy of entanglement' is not clearly explained. Considering this is not a commonly used concept in the disentangled representation learning community, I would suggest the authors provide more explanations on it. This may include (1) an intuitive descriptive introduction about what 'entropy of entanglement' is; (2) a detailed explanation on its relation to the disentangled representation learning; (3) a few experiments conducted on the disentanglement learning datasets to validate and demonstrate the claimed connections to 'entropy of entanglement'; (4) why it can solve the unidentifiability problem (Locatello et al., 2019); (5) how this 'entropy of entanglement' motivates the use of the tensor product of n unit circles as the representation.
2. The 4-th paragraph in the Sec. introduction says the n-torus $T^n$ has a low entanglement property. I wonder do the commonly used n-dim vector representations have low or high entanglement property? Which type is higher? How to compare these two types of representations quantitatively by 'entanglement property'?
3. I can see there are two steps of new techniques incorporated in the proposed method. One is the usage of multiple cyclic representations with each circle indexed by two 1-dim latent variables (defined by Eq. (6)(7)(8)). The other one is the flattening operation defined by the tensor product $\otimes$ in Eq. (5). Can the authors explain why they are motivated individually? More specifically, (1) why n unit circles are better than traditional vectors and (2) why we must use the tensor product to forward them to the decoder?
4. As the authors have noticed by themselves, the proposed tensor product method cannot generalize to a large number of latent dimensions because the dimensionality of the representation increases exponentially with the torus dimension. I think the authors should provide a solution to this obvious problem in this paper with experimental support.
5. No ablation study on the number of torus dimensions is provided in the experiment section.
6. Experiments on some complex datasets like CelebA should be provided to show the generalization ability of this method.

**Summary Of The Paper:**

This paper proposes to use the tensor product of n unit circles to learn the disentangled representation. Experiments on synthetic datasets show the effectiveness of this technique.

**Summary Of The Review:**

Based on my concerns, I think this paper requires some significant improvements and I rate it as rejected for now.

---

> ### Author Response · Authors · 2021-11-09
> **Ablation Study**
>
> Thank you for your comprehensive feedback. We will upload a detailed reply covering all of the comments in the following days.
>
> Since time is of the essence, it is a bit more urgent for us to understand the fifth comment: “No ablation study on the number of torus dimensions is provided in the experiment section.”. We believe that this issue is addressed in the Experiments section under “Dependence on $\beta$ and $D$”, and in Figure 4 where the DC-score, MSE, and FID are plotted as a function of the torus’ dimension, $D$.
>
> If this does not fully satisfy your concern, we would be grateful if you would elaborate on the type of ablation required and we would make an effort to comply ASAP.

---

> > ### Comment · Reviewer_UzjD · 2021-11-12
> > **Apology for the confusing comment**
> >
> > I'm sorry for this confusing comment.
> >
> > When I was writing the comments, another paper I read recently occurred to me, and I mistook your model as using discrete cyclic structures for a moment so I wondered how the number of predefined dims in a torus affects the experiments.
> > It's my fault for this question, please ignore it.
> >
> > Thanks.

---

> ### Author Response · Authors · 2021-11-20
> **Author Response**
>
> Thank you for your comprehensive feedback.
>
> Q: “The motivation 'entropy of entanglement' is not clearly explained. Considering this is not a commonly used concept in the disentangled representation learning community, I would suggest the authors provide more explanations on it. This may include (1) an intuitive descriptive introduction about what 'entropy of entanglement' is; (2) a detailed explanation on its relation to the disentangled representation learning; (3) a few experiments conducted on the disentanglement learning datasets to validate and demonstrate the claimed connections to 'entropy of entanglement'; (4) why it can solve the unidentifiability problem (Locatello et al., 2019); (5) how this 'entropy of entanglement' motivates the use of the tensor product of n unit circles as the representation.”
>
> A: 1. In addition to the explanatory paragraph in the introduction, we added an intuitive description of the concept of entanglement entropy as a measure of how far the state is from being a product state. The measure relies on the linear structure of quantum mechanics which is relevant to us as we construct a linear relation between the generative factors and the latent codes.
> 2. In the added appendix, we provide more details on the relation between the mathematical structure of tensor product representation used in the paper and the concept of disentanglement.
> 3. As shown in the appendix, the DC score has a $-P \log P$ term which is the analog of entanglement entropy. Therefore, all the experimental results that we present and have a high DC score validate the relation to low entanglement entropy.
> 4. As noted in the discussion section, the solution to the unidentifiability problem relies on the fact that we use as the inductive bias a specific choice of low entropy representation of the latent codes. This representation is a very non-generic state in the Hilbert space of states.
> 5. As noted in the introduction, in the discussion section and in the added appendix (section B), the low entanglement is a consequence of the tensor product structure of the torus.
>
> Q: “The 4-th paragraph in the Sec. introduction says the n-torus has a low entanglement property. I wonder do the commonly used n-dim vector representations have low or high entanglement property? Which type is higher? How to compare these two types of representations quantitatively by 'entanglement property'? I can see there are two steps of new techniques incorporated in the proposed method. One is the usage of multiple cyclic representations with each circle indexed by two 1-dim latent variables (defined by Eq. (6)(7)(8)). The other one is the flattening operation defined by the tensor product”
>
> A: The entanglement entropy is a measure of how far the state is from a product state. A generic vector in the Hilbert space is highly entangled. Only measure zero of the vectors in the Hlibert space take the form that we use and are low entangled. The flattening operation is a technical step that does not affect the low entanglement structure and occurs after the tensor product.
>
> Q: “in Eq. (5). Can the authors explain why they are motivated individually? More specifically, (1) why n unit circles are better than traditional vectors and (2) why we must use the tensor product to forward them to the decoder?”
>
> A: By definition, not all the vectors in Hilbert space have identical entanglement structure as measured by the entanglement entropy. Correlations between the different components of the vectors cause the entanglement entropy to increase. The structure that we use has by definition low entanglement since there are no correlations between the different components of the vector representation.
>
> Q: “As the authors have noticed by themselves, the proposed tensor product method cannot generalize to a large number of latent dimensions because the dimensionality of the representation increases exponentially with the torus dimension. I think the authors should provide a solution to this obvious problem in this paper with experimental support.”
>
> A: One expects that the majority of structured physical distributions that arise in practice can be captured by a small number of generative factors. When this is not the case one can still use a representation based on direct sums of tensor products, which can still be made low entangled.
>
> Q: “Experiments on some complex datasets like CelebA should be provided to show the generalization ability of this method.”
>
> A: The CelebA dataset contains only binary generative factors. Our method works well for continuous generative factors: both Lasso and the DC score measure are more appropriate for continuous factors than for binary ones.

---

> > ### Comment · Reviewer_UzjD · 2021-11-22
> > **About CelebA dataset**
> >
> > I did not mean to measure the entanglement quantitatively on CelebA but show qualitative results. The factors like azimuth change, smile change, haircolor change, etc, are continuous and should be learned by the model if it works. Latent traversals would be fine to show if the method works.

---

> > > ### Author Response · Authors · 2021-11-27
> > > **Author Response**
> > >
> > > The structure of human faces is more compositional (see portrait sketching software) than a product of continuous factors. However, the pose is composed of multiple continuous factors that our method should be able to recover given the appropriate dataset. While CelebA is said to cover large pose variations, the pose distribution is spread mostly around frontal or 30 degree images, since these are the types that are common on the internet.  We, therefore, employ the cropped Multi-PIE dataset  (https://github.com/bluer555/CR-GAN/blob/master/README.md#dataset), which contains 250 people and 9 predefined poses spread evenly on a ±60 degrees with two facial expressions (the original Mutli-PIE dataset is not freely available). As can be seen in the anonymous link https://imgur.com/a/6ek1F4B, the latent factors obtained are indeed indicative of lighting, azimuth, and yaw (no roll in the dataset).

---

> > ### Comment · Reviewer_UzjD · 2021-11-22
> > **About the Explanation of Quantum Physics**
> >
> > I appreciate the authors provide more explanations on quantum physics and entropy of entanglement, which are informative.
> >
> > However, from the additional explanation, I can only see that the product of cyclic structure seems to be of low entanglement in terms of quantum physics by the **structure definition**, but it cannot solve the problem of why the learned representation should be disentangled. Considering the existing disentanglement models, they all use a **grid structure** to learn the representations, and we all can see the **grid** seems to be intuitively disentangled by the structure definition itself, but it is not related to whether the learned representation is disentangled. I feel the same situation exists in the torus case, where we all know torus is a disentangled structure of cyclic factors, but why the learned representation should follow the disentanglement expectation is not explained.

---

> > > ### Author Response · Authors · 2021-11-27
> > > **Author Response**
> > >
> > > In the outer product torus representation we are adding an additional structure to the vector space by writing it as a tensor product of 2-vector spaces.
> > > This is essentially a choice of basis on the vector space. The low entangled representation is measured with respect to this basis. A generic vector will not have a low entanglement with respect to this choice of basis. Entanglement entropy is a basis dependent observable (by this we mean a global basis on the Hilbert space $H$ and not on the different tensor product factors $F$). A grid is a vector space (over the integers). It is not an outer product of one-dimensional vectors, but rather a direct product (see additional explanation below). Unless we introduce the above outer product structure, there is no meaning to low or high entanglement to its representation. Example:  let $H=F\otimes F$ be the Hilbert space of a two-state system. A state of the form $|0\rangle |1\rangle \in H$ is zero entangled vector while a state $\frac{|0\rangle |0\rangle + |1\rangle |1\rangle}{\sqrt{2}} \in H$ (Bell state) is maximally entangled. We stress that entanglement is a property of the linear algebra structure in general and not only of quantum mechanics.
> > >
> > > Note also that there is a fundamental difference between the interval and the circle that affects the outer (tensor) product structure of these objects. The circle has a non-trivial topology, hence its embedding in flat space has an extrinsic curvature. For this reason, in this case, the basic space from which we construct the outer product is two-dimensional despite the fact that the circle is one-dimensional. The dimension of the latent space is $2^D$. It is the same structure used in quantum mechanics where the qubit Hilbert space is two-dimensional and it leads to the low entangled structure.
> > >
> > > This is not the same with the interval, where the basic embedding space that is needed is one-dimensional. In this case, the latent space is a tensor product of one-dimensional vectors (scalars), hence we will be mapping each instance to a number (and not $2^D$-dimensional vector), and there is no sense in which we can define low entangled representations.

---

> > > > ### Comment · Reviewer_UzjD · 2021-11-29
> > > > **Why tensor product**
> > > >
> > > > Why tensor product? The inspiration of 'entanglement entropy' seems can only motivate the torus structure which is of low entanglement entropy, but why tensor product is used together with the torus structure? Is this tensor product essential? What's the performance with only the torus structure but without tensor product? What's the performance with only tensor product but without the torus structure (for example, we can remove the normalization Eq. (6) but still use a 2-dim basic space to obtain a $2^n$ outer product representation)? This ablation study is the key to determining which component is the actual part that brings the performance gain in this method.
> > > >
> > > > I feel this statement 'It is the same structure used in quantum mechanics where the qubit Hilbert space is two-dimensional and it leads to the low entangled structure.' is inspiring, and I personally do believe there are some interesting connections between quantum mechanics and disentanglement learning. However, I think this paper could be more convincing if the relation between disentanglement learning and quantum mechanics is more deeply investigated, and I still feel this paper in its current state does not provide enough insight or the insight is not convincingly explained.

---

> > > > > ### Author Response · Authors · 2021-11-29
> > > > > **Author Response**
> > > > >
> > > > > Q: “Why tensor product? The inspiration of 'entanglement entropy' seems can only motivate the torus structure which is of low entanglement entropy, but why tensor product is used together with the torus structure? Is this tensor product essential?
> > > > >
> > > > > A: The tensor product is essential since, as explained in detail in the appendix, by construction the entanglement entropy requires the tensor product structure in order to be able to define it. Without the tensor product structure, there is no sense in which we can define entanglement entropy and compare low and high entanglement.
> > > > >
> > > > > Q: "What's the performance with only the torus structure but without tensor product?"
> > > > >
> > > > > A: Training and running the VAE with $T^8$ without the tensor product structure (namely a direct product of the $m$’s) on the teapots dataset we get the scores below. The DC-score, disentanglement and completeness scores of the model without the tensor product structure are inferior compared to our results (see Table 2), while the MSE and FID are comparable to our method. This is expected since the latent codes are no longer disentangled.
> > > > >
> > > > > |Method  | DC-score        | D    | C     | I             | MSE           | FID         |
> > > > > |:----------------|:----------------|:----------------|:----------------| :----------------| :----------------| :----------------|
> > > > > | w/o tensor product |     0.3999    |   0.3709     |  0.4313    | 0.0658 | 0.0013 |  22.2275 |
> > > > > |Ours  |    **0.6230** | **0.6558** | **0.5919** | **0.0332** | **0.0012** |**15.7170**|
> > > > >
> > > > >
> > > > > Q: "What's the performance with only tensor product but without the torus structure (for example, we can remove the normalization Eq. (6) but still use a 2-dim basic space to obtain a outer product representation)? "
> > > > >
> > > > > A: In such a case we will be replacing the one-dimensional circle $S^1$ with a two-dimensional plane $R^2$. Our basic assumption is that each property is parametrized by a one-dimensional curve and not a two-dimensional plane. If we attempt to incorporate two properties with the two coordinates of the plane $R^2$ we will lose the tensor product structure since $R^2 \otimes R^2$ is a mixture of direct products ($R^2$ is a direct product of two $R$’s) and tensor products and we will not be able to disentangle the generative factors in each $R^2$ plane. Note, that there is no proper definition of entanglement associated with the direct product structure. Please also note that as explained in our previous reply on the hypercube, it is not possible to associate a property with a line or interval since the tensor product structure will be trivial.
> > > > >
> > > > > Q: “I feel this statement 'It is the same structure used in quantum mechanics where the qubit Hilbert space is two-dimensional and it leads to the low entangled structure.' is inspiring, and I personally do believe there are some interesting connections between quantum mechanics and disentanglement learning. However, I think this paper could be more convincing if the relation between disentanglement learning and quantum mechanics is more deeply investigated, and I still feel this paper in its current state does not provide enough insight or the insight is not convincingly explained.”
> > > > >
> > > > > A: Our construction parallels that one in quantum mechanics. In quantum mechanics one associates a two-dimensional vector space to each qubit and we do the same with each latent code. In quantum mechanics, one constructs the tensor product of the single-qubit Hilbert spaces in order to construct the Hilbert space of the quantum register. We do the same in order to describe the space of latent codes. In quantum mechanics one defines the entanglement entropy with respect to the tensor product decomposition. We do the same in our construction. The difference compared to quantum mechanics is that in quantum mechanics a qubit is described by two angles while a latent code in our case is parametrized by one angle. This follows from the fact that in quantum mechanics one works with linear vector spaces over the field of complex numbers, while we work with vector spaces over the field of real numbers.

---

> > ### Comment · Reviewer_UzjD · 2021-11-22
> > **About the Flattening Operation**
> >
> > By **flattening operation**, do the author mean the outer product operation and the appending orientation vector operation?
> >
> > The authors say this is a technical step and it is not related to the choice of low entanglement structure. But I wonder how this technical step affects the efficacy of the model, and why this technical step is motivated.
> >
> > If this technical step is very important for the overall efficacy of this model, does it mean the torus structure is not actually important? Can this technical flattening step be used independently by replacing the torus latent structure with traditional vectors?

---

> > > ### Author Response · Authors · 2021-11-27
> > > **Author Response**
> > >
> > > The flattening (denoted as vec in equation 5) operation is taking a tensor product which is an n-rank tensor, and writing it as a vector. In the example in figure 1, it simply means writing a 2 by 2 matrix as a 4-vector. It is explained in detail with an example below equation 5.
> > >
> > > After flattening the tensor, $v_{orient}$ is appended to it. It is required, since otherwise, the mapping of $v_{prod}$ would not be an embedding, i.e. will not be injective and therefore not a topological homeomorphism between the domain and its torus image. This is explained in the paragraph under equation 5.

---

### Official Review · Reviewer_YGh8 · 2021-11-01

**Correctness:** 4
**Technical Novelty And Significance:** 4
**Empirical Novelty And Significance:** 4
**Recommendation:** 8
**Confidence:** 3

**Main Review:**

Strengths
 - Method is simple yet innovative
 - The intuition behind the method is explained
 - Integrates very well with previous frameworks in the literature
 - Evaluation is convincing, including a direct demonstration of a strong linear match between ground truth and learned representations
 - Discussion of results satisfactorily accounts for why the method performs relatively worse in one dataset
 - The discussion frankly acknowledges a limitation of the method in terms of dealing with large numbers of latent causes

Weaknesses
 - Evaluation is based on artificial datasets, similar to the rest of the literature.  An attempt to use the method on real data, even if inconclusive, would be informative, especially since one of the limitations of the method is in dealing with large number of latent causes (which would presumably be much higher in real data than for simulations).
 - The explanations of the success of the method in terms of quantum physics concepts fall short of giving actual insight into why the method works.  Why are the physics analogies appropriate in this problem, if they are?
 - It is unclear whether the torus topology is as crucial as claimed.  If we ignore the circular structure, the method greatly resembles basis expansion methods used in linear models with higher-order interactions (https://en.wikipedia.org/wiki/Interaction_(statistics)).  Is nonlinear basis expansion the real secret ingredient to making this approach work, or does the torus topology (transforming angles into points on a 2d circle) play an additional role?
 - Is Lasso appropriate for evaluating toroidal representations, given the possibility of rotation?
 - How were the tuning parameters (beta, etc.) tuned?
 - I am baffled by how the learned factors can represent z2, z3, and z4 linearly in Figure 2, given the rotational invariance inherent in the toroidal representation.  Is it because of R, G, B are mostly represented in the linear part of the basis expansion v_orient?

Minor comments
 - The convention (as well as the rules of English) seems to have "autoencoder" as a single word, not two, but I could be mistaken.
 - In eq. 9, "N(0,1)" should be "N(0, sigma)"?

**Summary Of The Paper:**

This paper proposes a new autoencoder architecture that achieves better recovery of the latent factor structure in artificial datasets according to previously established metrics: disentanglement, completeness, and informativeness.  The key idea is to nonlinearly project a small number of latent factors into a higher-dimensional space, based on the topology of the torus, in such a way that prevents an arbitrary linear rotation of the latent factors from yielding an equivalent representation.  The approach is compared against existing approaches for variational autoencoders focused on learning disentangled representation, and achieves higher disentanglement and completeness, as well as competitive or best informativeness (reconstruction) on a majority of datasets considered.

**Summary Of The Review:**

This is an innovative method that looks very promising given the limitations of the evaluation on artificial data.  The paper is not perfect, and could improve in being more critical of the approach proposed, as well as reproducibility in terms of describing the evaluation procedure.  But the results clearly show an advantage over previous approaches on a variety of artificial datasets, which is technically impressive even if the practical impact of the method on real data is still unknown.

---

> ### Author Response · Authors · 2021-11-20
> **Author Response**
>
> Thank you for your supportive review.
>
> Q: “Evaluation is based on artificial datasets, similar to the rest of the literature. An attempt to use the method on real data, even if inconclusive, would be informative, especially since one of the limitations of the method is in dealing with large number of latent causes (which would presumably be much higher in real data than for simulations).”
>
> A:  The reviewer raises an important issue for future studies. We note that a major complexity when dealing with real data is that it lacks annotation of the generative factors.
>
> Q: “The explanations of the success of the method in terms of quantum physics concepts fall short of giving actual insight into why the method works. Why are the physics analogies appropriate in this problem, if they are?”
>
> A:  We added an  explanation to the appendix that covers the concepts of quantum physics and relationship to this work:
> The assumption is there is a set of elementary generative factors $z_i,i=1,..,n$ that are disentangled from each other, and from which the data distribution can be constructed. The task is to learn these elementary factors and represent them by latent codes $c_a,a=1,...,m$. Ideally we would have liked to have $n=m$ and a bijective map $f$ that associates to any $c_a$ one $z_i$ and vice versa.
>
> We are working with a linear representation and since quantum mechanics is a linear theory it inspires our choice of the latent space representation. In quantum mechanics states are represented by vectors in Hilbert space and the operators that act on them are linear. In our case, the generative factors constitute a state and so do the latent codes. The linear regressor relating them is a linear operator.
>
> We represent each elementary factor $z_i$ and a state (qubit) in a Hilbert space $|z_i\rangle \in {\cal F}$ and the complete set of elementary factor by a state (quantum register of $n$ qubits) in the tensor product of these Hilbert spaces  $|z_1,...,z_n\rangle  = |z_1\rangle\cdot\cdot\cdot |z_n\rangle \in {\cal H} = F\otimes F\cdot\cdot\cdot \otimes F$.
> Note, that only a measure zero of $n$-qubit states can be represented in such a way as a product state and the reason that we can do that is the assumption that $z_i$ are disentangled.
>
> Entanglement can be viewed as a measure of how far the state is from being a product state.
> One way to measure it is by taking the supremum on the absolute value
> of the inner product of the state with all the possible product states.
> Another measure is the Von-Neumann entropy that is calculated by constructing a density matrix $\rho$ as the outer product of $|z_1,...,z_n\rangle$, decomposing
> $z_i$ to e.g. two subsets $A$ and $B$ such that $\rho \in {\cal H}^A\otimes
> {\cal H}^B$ and tracing out part $B$, $\rho^A = Tr_B \rho$. The eigenvalues of $\rho^A$
> encode the entanglement between $A$ and $B$. Specifically, the Von-Neumann entropy
> is $S_{VN} = - Tr \rho^A \log \rho^A$ and it vanishes iff the state $|z_1,...,z_n\rangle$ is a product state.
>
> Returning to our case, we construct the latent space representation using a tensor product
> since it should capture the above structure of the generative factors. The measure of disentanglement ${\cal D}_a$ quantifies it by the $-P \log P$ term which is the analog of the Von-Neumann entropy.
>
> Note that there is a difference between our case and quantum mechanics: in quantum mechanics an elementary qubit state is represented by two angles $(\theta,\phi)$ while in our case each elementary latent code is represented by one angle parametrizing a circle of the torus.

---

> ### Author Response · Authors · 2021-11-21
> **Author Response #2**
>
> Q: “It is unclear whether the torus topology is as crucial as claimed. If we ignore the circular structure, the method greatly resembles basis expansion methods used in linear models with higher-order interactions (https://en.wikipedia.org/wiki/Interaction_(statistics)). Is nonlinear basis expansion the real secret ingredient to making this approach work, or does the torus topology (transforming angles into points on a 2d circle) play an additional role?
>
> A: One may inquire what would have gone wrong in our analysis if we chose that space of latent variables to be another curved compact manifold such as a sphere. In such a case we should have covered the manifold with local patches such that there would be a linear map between the generative factors and the latent variables on each patch and a transition function between them. Such a representation would be entangled and will fail, e.g. on the sphere near the pole since rotating by the azimuthal angle would not change any property.
>
> Q: “Is Lasso appropriate for evaluating toroidal representations, given the possibility of rotation?”
>
> A: The Lasso is still appropriate for evaluating toroidal representations since as can be seen from the heatmap in Figure 2, when the generative factor is not periodic the corresponding latent variable takes values in an interval instead of the full circle.
>
> Q: “How were the tuning parameters (beta, etc.) tuned?”
>
> A: By default, we chose $\beta=1$ as in the original VAE. We verified this selection in the experiments section under “Dependence on $\beta$ and D”.
>
> Q: “I am baffled by how the learned factors can represent z2, z3, and z4 linearly in Figure 2, given the rotational invariance inherent in the toroidal representation. Is it because R, G, B are mostly represented in the linear part of the basis expansion v_orient?”
>
> A: The RGB color values are indeed non-periodic in nature. Please note that the values of the codes, $\theta$, do not cover the whole circle in this case, and in fact, only map a subset of the circle’s domain to the RGB values, thus locally, the codes for the RGB values are non-periodic as well.
>
> Thank you for pointing out the typos, which were corrected in the revised manuscript.

---

> > ### Comment · Reviewer_YGh8 · 2021-11-25
> > **How about a hypercube?**
> >
> > Thanks for your replies.  I have one more question following your explanation:
> >
> > "One may inquire what would have gone wrong in our analysis if we chose that space of latent variables to be another curved compact manifold such as a sphere. In such a case we should have covered the manifold with local patches such that there would be a linear map between the generative factors and the latent variables on each patch and a transition function between them. Such a representation would be entangled and will fail, e.g. on the sphere near the pole since rotating by the azimuthal angle would not change any property."
> >
> > I accept that using a sphere would be difficult, but what about a hypercube? I.e. letting the latent space variables be coordinates restricted to [0,1].  That space is compact, and does not seem to introduce any problems with the entanglement.

---

> > > ### Author Response · Authors · 2021-11-27
> > > **Author Response**
> > >
> > > Thank you for the question. There is a fundamental difference between the interval and the circle that affects the outer (tensor) product structure of these objects. The circle has a non-trivial topology, hence its embedding in flat space has an extrinsic curvature. For this reason, in this case, the basic space from which we construct the outer product is two-dimensional despite the fact that the circle is one-dimensional. The dimension of the latent space is $2^D$. It is the same structure used in quantum mechanics where the qubit Hilbert space is two-dimensional and it leads to the low entangled structure.
> > >
> > > This is not the same with the interval, where the basic embedding space that is needed is one-dimensional. In this case, the latent space is a tensor product of one-dimensional vectors (scalars), hence we will be mapping each instance to a number (and not $2^D$-dimensional vector), and there is no sense in which we can define low entangled representations.

---

> > > > ### Comment · Reviewer_YGh8 · 2021-11-29
> > > > **That helps**
> > > >
> > > > Thank you for your explanation.  Similar to other reviewers, I'm haven't studied the mathematics of quantum mechanics comprehensively, so if you could unpack some of the QM concepts in terms of the more fundamental mathematical ideas (and I know that could be asking a lot!), that would be helpful.
> > > >
> > > > I can see that one difference between a cube and a torus is that you could rotate a cube and map it back to the original cube while preserving the local geometry in most places.  With a torus, it seems less possible to "rotate" the coordinates in a way that can be smoothly mapped back to the original space.  Does that intuition capture some of the reasoning behind why a torus, but not a hypercube, can faithfully represent a low-entanglement latent space?

---

> > > > > ### Author Response · Authors · 2021-11-30
> > > > > **Author Response #1**
> > > > >
> > > > > Q: Thank you for your explanation. Similar to other reviewers, I'm haven't studied the mathematics of quantum mechanics comprehensively, so if you could unpack some of the QM concepts in terms of the more fundamental mathematical ideas (and I know that could be asking a lot!), that would be helpful.
> > > > >
> > > > > A: As an example of a quantum system consider the quantum computer. In the following, we will attempt to briefly describe this quantum system and relate the concepts to the ones that we use in our work.
> > > > > 1. The basic element is the quantum bit, qubit, that generalizes the classical bit. While a classical bit takes values $0$ or $1$, the qubit can be in a linear superposition of $0$ and $1$. One represents the qubit state $|\psi\rangle$ as a vector in a two-dimensional Hilbert space $H=C^2$ that can be expanded in the basis $[|0\rangle, |1\rangle]$ as $|\psi\rangle = \alpha |0\rangle + \beta |1\rangle$ where $\alpha$ and $\beta$ are complex numbers that satisfy the normalization condition $|\alpha|^2 + |\beta|^2 = 1$. One has the equivalence relation between quantum states $|\psi\rangle \sim e^{i\phi} |\psi\rangle$, thus a qubit state is described by two real variables (one complex variable). This can be parametrized by two angles $\theta,\varphi$ that parametrize a sphere (Bloch sphere) $S^2 \simeq CP^1$, where $CP^1$ is the complex projective space.
> > > > > In our case, we replace the two-dimensional space  $H=C^2$ by $R^2$. The qubit is the analog of our latent code. However, instead of the two angles needed for the qubit parametrization, we need one angle to parametrize the latent code. This follows from the fact that in quantum mechanics one works with linear vector spaces over the field of complex numbers, while we work with vector spaces over the field of real numbers. Note, that we assume that each generative factor is parametrized by one real number.
> > > > > 2. A quantum register consists of $D$ qubits whose state is a vector in the Hilbert space ${\cal H} = C^2 \otimes C^2\cdot\cdot\cdot C^2$ (tensor product space). Its complex dimension is $dim {\cal H }= 2^D$. One can choose a basis for this Hilbert and expand the $D$-qubit state in it. Similarly to the qubit, the normalization condition and the equivalence phase relation imply that the general $D$-qubit state can be parametrized by the $CP^{2^D-1}$ - the complex projective space. A special $D$-qubit disentangled state (that we will discuss in item (3) below)  is parametrized by $S^2 \otimes S^2\cdot\cdot\cdot S^2 = (S^2)^D$.
> > > > > The $D$-qubit quantum register is the analog of latent space (the $D$ latent codes). The $D$-qubit Hilbert space is the analog of the tensor product space $R^2\otimes R^2\cdot\cdot\cdot R^2 = (R^2)^D$ in our work, whose real dimension is $2^D$. The special $D$-qubit space $(S^2)^D$ is the analog of our torus $(S^1)^D$.
> > > > > 3. A generic $D$-qubit state $|\psi\rangle$ cannot be written as tensor product state, i.e. $|\psi_1,...,\psi_D\rangle \neq {|\psi_1\rangle}_1 \otimes {|\psi_2\rangle}_2 \cdot\cdot\cdot {|{\psi}_D\rangle}_D$ where $|\psi_i\rangle_i \in H_i$. When it can be written as a tensor product, it means that there is no interaction between the qubits and their properties (like spin, momentum etc.) are not correlated. This is the special family of states in item (2).
> > > > > In our case, a generic vector representing the latent codes in the tensor product space $(R^2)^D$ cannot be written as a tensor product of vectors in each $R^2$. The special family that can be written like that is parametrized by the torus $T^D$ and this is the one that we choose.
> > > > > 4. The entanglement entropy is a measure of how the $D$-qubits are entangled. It is zero when there is no interaction between them (product state) and its maximal value is $D$ (in Log basis $2$). Entanglement can be viewed as a measure of how far the state is from being a product state. A measure of entanglement is the Von-Neumann entropy that is calculated by constructing a density matrix $\rho$ as the outer product of $|\psi_1,...,\psi_D\rangle$, decomposing $\psi_i$ to e.g. two subsets $A$ and $B$ such that $\rho \in {\cal H}^A\otimes {\cal H}^B$ and tracing out part $B$, $\rho^A = Tr_B \rho$. The eigenvalues of $\rho^A$ encode the entanglement between $A$ and $B$. Specifically, the Von-Neumann entropy is $S_{VN} = - Tr \rho^A \log \rho^A$ and it vanishes iff the state $|\psi_1,...,\psi_D\rangle$ is a product state.
> > > > > In our case, the measure of disentanglement ${\cal D}_a$ quantifies it by the $-P \log P$
> > > > > term which is the analog of the Von-Neumann entropy.
> > > > >
> > > > >
> > > > > We will add this explanation as a background section in the appendix.

---

> > > > > > ### Author Response · Authors · 2021-11-30
> > > > > > **Author Response #2**
> > > > > >
> > > > > > Q: I can see that one difference between a cube and a torus is that you could rotate a cube and map it back to the original cube while preserving the local geometry in most places. With a torus, it seems less possible to "rotate" the coordinates in a way that can be smoothly mapped back to the original space. Does that intuition capture some of the reasoning behind why a torus, but not a hypercube, can faithfully represent a low-entanglement latent space?
> > > > > >
> > > > > > A: We do not see a direct relation between the global rotation of the whole geometrical structure in the embedding space and entanglement. Entanglement is related to the correlation between the different components that constitute the global structure. In quantum mechanics, these are the $S^2$’s that parametrize the qubits and in our case these are the $S^1$’s that parametrize the latent codes. In this regard, the hypercube does not have a tensor product structure of its basic one-dimensional elements.

---

### Official Review · Reviewer_4Dgs · 2021-11-02

**Correctness:** 3
**Technical Novelty And Significance:** 4
**Empirical Novelty And Significance:** 3
**Recommendation:** 6
**Confidence:** 3

**Main Review:**

This study provides interesting results when using the torus as the representation space. I think the idea is novel. The utility is evaluated 5 datasets, and the results are convincing. I enjoyed reading this paper.

The main drawback of this paper is the lack of theoretical analysis. While the motivation of introducing the torus is based on theory, the property of the proposed latent representation (5) is not analyzed. For example, could you provide a lower bound of DC score when we employ (5)?

I'm puzzled about the connection between this study and quantum physics. I'm not an expert of quantum physics, and I have no idea what are entanglement property, Von Neumann entropy, and so on. Please elaborate on these concepts and how they are related to the torus representation. I believe this revision significantly enhances the manuscript.

Minor comments:
- Typo: n in (5) seems to be D.
- The sentence below (8): the index (a, \alpha) should be an integer, but why they are distributed from the Gaussian?

**Summary Of The Paper:**

The authors propose an autoencoder where the latent space is defined on a torus. A D-dimensional torus is represented by the tensor product of D unit circles. They argue the torus induces disentanglement as the analogous of entropy entanglement in quantum physics. Using 5 datasets, they show both the reconstruction performance and disentanglement score (Eastwood & Williams, 2018) are high.

**Summary Of The Review:**

The concept of the torus representation is novel and its practical value is empirically convincing. However, there is no assessment in terms of theory. The connection to quantum physics is not well explained.

---

> ### Author Response · Authors · 2021-11-20
> **Author Response**
>
> Thank you for your mostly supportive review.
>
> Q: “The main drawback of this paper is the lack of theoretical analysis. While the motivation of introducing the torus is based on theory, the property of the proposed latent representation (5) is not analyzed. For example, could you provide a lower bound of DC score when we employ (5)?”
>
> A: The main source of error is a misidentification of the number of generative factors ($K$) and having a much smaller number of latent codes ($D$). In such a case, the ratio $\frac{rank(R)}{K}$ in equation (1) is small making the disentanglement and hence the DC score small.
>
> Q : “I'm puzzled about the connection between this study and quantum physics. I'm not an expert of quantum physics, and I have no idea what are entanglement property, Von Neumann entropy, and so on. Please elaborate on these concepts and how they are related to the torus representation. I believe this revision significantly enhances the manuscript.”
>
> A:  We added an  explanation to the appendix that covers the concepts of quantum physics and relationship to this work:
> The assumption is there is a set of elementary generative factors $z_i,i=1,..,n$ that are disentangled from each other, and from which the data distribution can be constructed. The task is to learn these elementary factors and represent them by latent codes $c_a,a=1,...,m$. Ideally we would have liked to have $n=m$ and a bijective map $f$ that associates to any $c_a$ one $z_i$ and vice versa.
>
> We are working with a linear representation and since quantum mechanics is a linear theory it inspires our choice of the latent space representation. In quantum mechanics states are represented by vectors in Hilbert space and the operators that act on them are linear. In our case, the generative factors constitute a state and so do the latent codes. The linear regressor relating them is a linear operator.
>
> We represent each elementary factor $z_i$ and a state (qubit) in a Hilbert space $|z_i\rangle \in {\cal F}$ and the complete set of elementary factor by a state (quantum register of $n$ qubits) in the tensor product of these Hilbert spaces  $|z_1,...,z_n\rangle  = |z_1\rangle\cdot\cdot\cdot |z_n\rangle \in {\cal H} = F\otimes F\cdot\cdot\cdot \otimes F$.
> Note, that only a measure zero of $n$-qubit states can be represented in such a way as a product state and the reason that we can do that is the assumption that $z_i$ are disentangled.
>
> Entanglement can be viewed as a measure of how far the state is from being a product state.
> One way to measure it is by taking the supremum on the absolute value
> of the inner product of the state with all the possible product states.
> Another measure is the Von-Neumann entropy that is calculated by constructing a density matrix $\rho$ as the outer product of $|z_1,...,z_n\rangle$, decomposing
> $z_i$ to e.g. two subsets $A$ and $B$ such that $\rho \in {\cal H}^A\otimes
> {\cal H}^B$ and tracing out part $B$, $\rho^A = Tr_B \rho$. The eigenvalues of $\rho^A$
> encode the entanglement between $A$ and $B$. Specifically, the Von-Neumann entropy
> is $S_{VN} = - Tr \rho^A \log \rho^A$ and it vanishes iff the state $|z_1,...,z_n\rangle$ is a product state.
>
> Returning to our case, we construct the latent space representation using a tensor product
> since it should capture the above structure of the generative factors. The measure of disentanglement ${\cal D}_a$ quantifies it by the $-P \log P$ term which is the analog of the Von-Neumann entropy.
>
> Note that there is a difference between our case and quantum mechanics: in quantum mechanics an elementary qubit state is represented by two angles $(\theta,\phi)$ while in our case each elementary latent code is represented by one angle parametrizing a circle of the torus.
>
>
> One may inquire what would have gone wrong in our analysis if we chose that space of latent codes to be another curved compact manifold such as a sphere. In such a case we should have covered the manifold with local patches such that there would be a linear map between the generative factors and the latent codes on each patch and a transition function between them. Such a representation would be entangled and will fail, e.g. on the sphere near the pole since rotating by the azimuthal angle would not change any property.
>
>
> Minor comments:
> We corrected the typo in equation (5). We also corrected the index to component, as each component of the vector is sampled from the Gaussian distribution. Thank you for pointing these out.

---

### Official Review · Reviewer_DucY · 2021-11-02

**Correctness:** 4
**Technical Novelty And Significance:** 3
**Empirical Novelty And Significance:** 3
**Recommendation:** 8
**Confidence:** 4

**Main Review:**

Strengths: I enjoyed this paper. It is well written and tackles an important problem of disentanglement of latent factors in VAEs. The method of decomposing onto a set of circles that form a D-dimensional torus makes sense and is clearly described. The empirical tests are thorough, presented honestly, and support the theoretical intuition both qualitatively and quantitatively.  The figures are well done and aid in the flow. Figures 2 and 3 certainly demonstrate the effectiveness of the method!

Weaknesses: I struggled to find any major problems with this paper. My main unresolved questions about the method are: (1) how does it scale with the dimensions of the tori and (2) how it preforms when a factor is not well represented as a circle. These are addressed by the authors in the conclusion. I do not think (1) is a major weakness as I think the simplicity of the idea offsets a problem which may not arise in practice, this method will work well when the data are well explained by a few latent factors. Question (2) is central to the method, but I also think it raises a qualm that is beyond the scope of this paper. Since these were my main questions, I would like to see the heatmaps as in Figure 2 for the other experiments described in the “Dependence on beta and on D” section, but these could be added to an appendix. The reason is I would like to see the behaviour as D is varied as this would reveal more clearly how the method handles over and underspecification of the latent space.

I would also like to see a deeper investigation of the proposed DC-score metric, but, as is, it makes sense and is effective. Any further development of this element of the paper is likely beyond the scope.

Minor quibbles:
1) Equation (9) could be rewritten so it is less cramped and easier to parse.
2) The terms alpha_k in the same section only take on values 0 and 1, otherwise the result would not be a torus. Do the authors intend for this to be a future modelling choice point? If not, I think the notation could be cleaned up to improve clarity.



**Summary Of The Paper:**

The paper presents a novel method for disentangling latent representations, specifically for use with variational auto-encoders. This is accomplished by treating the latent codes as a tensor product of 1-spheres which results in their forming an n-dimensional torus where n is the number of latent factors. Additionally, the authors propose a new metric for analysing the quality of the decompositions. These methods are well founded and make intuitive sense, and the authors supplement this theoretical motivation with an array of tests comparing to several existing methods on several different datasets. The results demonstrate that the methods work well both qualitatively and quantitatively.

**Summary Of The Review:**

I think the paper is well composed, interesting, and convincing. Its methodology is well founded and is clearly communicated, and the empirical results strongly support method. I think it also opens many important directions for subsequent research. For these reasons, barring something major that I have overlooked, I recommend acceptance.

---

> ### Author Response · Authors · 2021-11-20
> **Author Response**
>
> Thank you for your supportive review.
>
> Q: “how it performs when a factor is not well represented as a circle”
>
> A: A continuous compact factor which naturally is not represented on the circle may still be represented by our method on a subset of the circle. Consider the representation of RGB colors in Figure 2: the generative factors of the colors are only mapped to codes in the range of [-2,2] and not from [-$\pi$,$\pi$].
>
> Q: “I would also like to see a deeper investigation of the proposed DC-score metric”
>
> A: We added a section to the appendix that clarifies the (quantum mechanical) intuition behind the components of the DC-score that favor a low entanglement entropy representation.
>
> Q: “I would like to see the heatmaps as in Figure 2 for the other experiments described in the “Dependence on beta and on D” section”
>
> A: We added additional heatmaps to the appendix as suggested.
>
> Minor quibbles:
>  We rewrote equation (9).
> The $\alpha_k$ terms can be used to implement higher-dimensional compact topologies including curved manifolds such as higher-dimensional spheres.

---

> > ### Comment · Reviewer_DucY · 2021-11-29
> > **Post-rebuttal Comments**
> >
> > Thank you for your revisions. I think the paper is improved with the additions in the appendix and the edits in the main text. I believe the paper should be accepted.

---

### Decision · Program_Chairs · 2022-01-20

**Decision:**

Accept (Poster)

**Comment:**

Three reviewers had a positive impression of this paper, two of them were willing to champion it. The main positive aspects mentioned by these reviewers were clarity, methodological strength, novelty and convincing experimental evaluation. On the other hand, the was one clearly negative vote, raising issues about the proposed concept of 'entropy of entanglement' and about the use of tensor products. It seems that after the rebuttal, this reviewer was still not fully convinced. In my opinion, however, the rebuttal addressed most of these points of criticism in a clear and transparent way, so I recommend acceptance.